# Comparing 3D Tooth Movement When Implementing the Same Virtual Setup on Different Software Packages

**DOI:** 10.3390/jcm11185351

**Published:** 2022-09-12

**Authors:** Azad Dhingra, Juan Martin Palomo, Neda Stefanovic, Manhal Eliliwi, Tarek Elshebiny

**Affiliations:** 1Department of Orthodontics, Case Western Reserve University, Cleveland, OH 44106, USA; 2Department of Orthodontics, University of Belgrade, 11000 Belgrade, Serbia

**Keywords:** orthodontics, virtual tooth movement, clear aligners

## Abstract

Background/objectives: The purpose of this study was to compare the differences in tooth movements when implementing the same virtual setup on the following four different software packages: ClinCheck^®^ Pro, Ortho Analyzer^®^, SureSmile^®^, and Ortho Insight 3D^®^. Materials/Methods: Twenty-five adult patients treated with Invisalign^®^ at the Case School of Dental Medicine (CWRU)’s department of orthodontics were retrospectively collected. Initial stereolithography (STL) files were obtained and imported into three software packages. The teeth were moved in order to replicate the virtual setup from ClinCheck^®^ Pro. The final outcomes were exported from each software package. ClinCheck^®^ Pro STL files were used as the reference while STL files produced by the other software packages were used as the targets. Best fit superimpositions were performed using Geomagic^®^ Control X. Based on the results, tooth position was adjusted in the three software packages until the virtual setups from ClinCheck^®^ Pro were replicated. Once confirmed, the tables containing the tooth movements were compared. The number of aligners and number of attachments automatically generated from each of the software packages were also evaluated. Results: Extrusion/intrusion (*p* ≤ 0.0001) and translation buccal/lingual (*p* ≤ 0.0004) were significantly different among the software packages. ClinCheck^®^ Pro and SureSmile^®^ (*p* ≤ 0.000), SureSmile^®^ and Ortho Insight 3D^®^ (*p* ≤ 0.014), SureSmile^®^ and Ortho Analyzer^®^ (*p* ≤ 0.009), and Ortho Insight 3D^®^ and Ortho Analyzer^®^ (*p* ≤ 0.000) generated a significantly different number of maxillary aligners. The results varied slightly for mandibular aligners, with only ClinCheck^®^ Pro and Ortho Insight 3D^®^ (*p* ≤ 0.000), SureSmile^®^ and Ortho Insight 3D^®^ (*p* ≤ 0.000), and Ortho Insight 3D^®^ and Ortho Analyzer^®^ (*p* ≤ 0.000) exhibiting a significant difference. ClinCheck^®^ Pro and SureSmile^®^ (*p* ≤ 0.000) differed significantly in the number of attachments produced. Conclusions: There are statistically significant differences in extrusion/intrusion, translation buccal/lingual, the number of aligners, and the number of attachments when implementing the same virtual setup on different software packages. Clinicians may need to consider this when utilizing software programs for digital diagnosis and treatment planning.

## 1. Introduction

Over the last century, several advancements have shaped the field of orthodontics and dentofacial orthopedics into what it is today. The concept of using light wires and intermaxillary elastics was introduced by Dr. Calvin Case in 1907 [1]. Appliances such as the E-arch, pin-and-tube, ribbon arch, and edgewise were released by Dr. Edward Angle in the 1920s [2]. The first self-ligating bracket was described by Dr. Jacob Stolzenberg in 1935 [3], and the straight-wire appliance—in which brackets were individually designed for each tooth—was proposed by Dr. Lawrence Andrews in 1976 [4]. However, few innovations, if any, have had as great of an impact as the introduction of clear aligners by Zia Chishti and Kelsey Wirth in 1997.

Although clear aligner therapy has become increasingly popular over the past decade, the concept of moving teeth without wires and brackets is not a new one. Dr. Harold Kesling released his “tooth positioning appliance” in 1945 [5]. The device was intended to be used as a finishing appliance after orthodontic treatment with full fixed appliances. If used as prescribed, it could correct rotations, establish proper interdigitation, and achieve good root parallelism. Kesling stressed the importance of a diagnostic wax setup during fabrication. Specifically, he described two key factors: clinicians must respect the biologic limitations of tooth movements and should only apply movements that are possible considering the available anchorage [5]. While Kesling was undoubtedly referring to conventional diagnostic setups in his statement, it is important for clinicians to be mindful of these factors when constructing virtual setups as well.

The development of computer-aided design began in the 1950s at the Massachusetts Institute of Technology [6]. Soon thereafter, computer-aided design and manufacturing (CAD/CAM) systems were incorporated into industrial planning and production. It took an additional thirty years, however, for CAD/CAM to be introduced to the field of dentistry by means of intraoral scanners (IOSs). IOSs were initially used in restorative dentistry to capture the direct optical impressions of limited areas. As the technology improved, IOSs became capable of capturing complete dental arches and, in 2008, Cadent released the first IOS suitable for orthodontics—the iTero [6,7]. Since then, many full arch IOSs have emerged on the market. 

It has been shown that digital orthodontic models have comparable accuracy and reliability to plaster models. They can therefore be considered an adequate replacement for clinical and research purposes [6,7,8]. IOSs have the benefit of enhancing patient comfort, clinical efficiency, and precision. Digital study models mean no discomfort related to the taking of impressions, readily available diagnostic records, and no storage issues. Not only do diagnoses and treatment planning become faster, they become more precise as well. Study model analysis, treatment simulations, indirect bonding procedures, digital appliance design, and surgical planning can all be accomplished using a digital workflow [8,9,10,11,12].

In 1997, two graduate students from Stanford University founded Align Technology, the company that initiated the use of CAD/CAM for the movement of teeth [13]. Invisalign was the first [14,15] and is currently the largest provider of custom-made clear aligners for orthodontic tooth movement [16]. The initial focus was on cases with mild crowding or spacing [17]. The Invisalign system has evolved over the years, however, and practitioners have acquired additional expertise using it. Treatment efficacy has improved substantially, and tooth movements such as rotation correction, incisor torquing, and molar distalization are more easily achieved. It is now possible to treat complex malocclusions and Invisalign, in consequence, has been deemed an esthetic alternative to fixed orthodontic appliances [18,19,20,21,22].

Recently, several companies have implemented CAD/CAM and embraced the concept of virtual tooth movement [23,24]. They have launched new software platforms to incorporate clear aligners as a treatment modality. While Invisalign only offers full-service aligner therapy and virtual setups, other companies have expanded on the idea and offer additional services including the digital fabrication of appliances, indirect bonding (IDB) trays, and in-house aligners [23,24].

Three-dimensional printing has infiltrated the orthodontic field and has become very popular [25]. More and more orthodontic applications are being fabricated using this technology. In order to print on three-dimensional printers, software needs to be used to create the necessary model so that applications can be fabricated [26]. There are numerous software programs that exist today that are capable of initiating innovative techniques in orthodontics, such as creating orthodontic study models, fabricating indirect bonding trays, and making models to fabricate in-house aligners. In the case of creating your own in-house aligner therapy, an orthodontist requires equipment in their armamentarium, including a digital scanner, aligner planning software, three-dimensional printer, and a thermoforming machine. There is a wide variety of aligner planning software that clinicians could utilize to virtually move teeth during diagnosis and treatment planning.

The purpose of this study was to compare the differences in tooth movements when implementing the same virtual setup on the following four different software packages: ClinCheck Pro, Ortho Analyzer, SureSmile, and Ortho Insight 3D.

## 2. Materials and Methods

This retrospective study was approved by the Case Western Reserve University Institutional Review Board (IRB STUDY20201221). Adult patients treated with Invisalign at the department of orthodontics were included in the study as long as their ClinChecks were from the SmartTrack era and, in turn, had tooth movement tables that were accessible. Patients with a history of facial trauma, craniofacial abnormalities, tooth malformation, intermaxillary elastic use, centric relation/centric occlusion (CR/CO) slide, or impacted teeth were excluded from the study. To achieve 90% power with an alpha value of 0.05, a sample size of twenty-four patients was deemed necessary. Twenty-five patients were consequently selected based on the specified criteria.

To begin, final Standard Tessellation Language (STL) files from the approved ClinCheck were exported from the ClinCheck Pro software (Align Technology, Santa Clara, CA, USA). Initial STL files were then obtained and imported into the following three software packages: Ortho Analyzer (3Shape, Copenhagen, Denmark), SureSmile (Dentsply Sirona, Charlotte, NC, USA), and Ortho Insight (MotionView software, Chattanooga, TN, USA). Here, the teeth were moved in order to replicate the virtual setup from ClinCheck Pro. Final outcomes were exported from each of the software packages and were saved as STL files. ClinCheck Pro STL files were used as the reference while STL files produced by the other software packages were used as targets. Best fit superimpositions were performed using Geomagic Control X (3D Systems, Rock Hill, SC, USA) and color-coded maps were obtained. 

Based on the results, tooth position was adjusted in the three software packages to ensure that the virtual setups from ClinCheck Pro were replicated (Figure 1). Only tooth surfaces were included in the superimpositions; gingiva and model bases were excluded. The tolerance for comparison was set from −0.25 mm to 0.25 mm in Geomagic Control X. Figure 2 illustrates superimpositions outside of the tolerance, indicating that the virtual setups differ from ClinCheck Pro. Figure 3, in contrast, shows superimpositions within the tolerance, indicating comparable results.

Once the virtual setups from the three software packages were confirmed to be equivalent to that of ClinCheck Pro, the tables containing the tooth movements’ extrusion/intrusion, translation buccal/lingual, translation mesial/distal, rotation mesial/distal, angulation mesial/distal, and inclination buccal/lingual were compared. The number of aligners automatically generated from each of the software packages was also evaluated. Furthermore, the software packages with an automatic generation of the attachments—specifically, ClinCheck Pro and SureSmile—were analyzed. The study design is outlined in Figure 4. To quantify the differences between the superimposed virtual setups, the negative average, positive average, absolute average, and standard deviation values were obtained. The inter-rate reliability was assessed by the intraclass correlation coefficient (ICC) based on 20% of the sample.

## 3. Statistical Analysis

All tests were conducted in statistical package for the social sciences (SPSS 26.0, IBM, Armonk, NY, USA) software. The Shapiro–Wilk normality test was performed, and it was determined that the variables were not normally distributed. Since non-parametric tests were indicated, the Kruskal–Wallis one-way analysis of variance was used for differentiation. Pairwise comparisons were implemented to evaluate the specific interaction between variables. *p* values of ≤0.05 were considered statistically significant.

## 4. Results

ICC showed a high degree of reliability for the repeated methodology (0.981). The Shapiro–Wilk normality test demonstrated that the variables were not normally distributed. The Kruskal–Wallis one-way analysis of variance showed that extrusion/intrusion (*p* ≤ 0.0001) and translation buccal/lingual (*p* ≤ 0.0004) were significantly different (Table 1) among the four software packages. The pairwise comparison showed that ClinCheck Pro and Ortho Insight 3D (*p* ≤ 0.026), ClinCheck Pro and Ortho Analyzer (*p* ≤ 0.000), SureSmile and Ortho Analyzer (*p* ≤ 0.000), and Ortho Insight 3D and Ortho Analyzer (*p* ≤ 0.000) all differed in the extrusion/intrusion direction (Table 2). Similarly, ClinCheck Pro and Ortho Insight 3D (*p* ≤ 0.004), ClinCheck Pro and Ortho Analyzer (*p* ≤ 0.001), SureSmile and Ortho Insight 3D (*p* ≤ 0.009), and SureSmile and Ortho Analyzer (*p* ≤ 0.002) differed significantly in their translation buccal/lingual movement values (Table 3). The four remaining movements—translation mesial/distal, rotation mesial/distal, angulation mesial/distal, and inclination buccal/lingual—did not differ significantly among the software packages (Table 1).

The equivalency of the virtual setups was confirmed by superimposing the models on Geomagic Control X. The maximum range was set from −1.0 mm to 1.0 mm, while the tolerance for comparison was set from −0.25 mm to 0.25 mm. The negative average, positive average, absolute average, and standard deviation values were obtained to quantify the differences between the virtual setups. The positive average was calculated to be 0.064 mm while the negative average was calculated to be −0.055 mm. The absolute average between the models was found to be 0.011 ± 0.086 mm (data provided in Supplemental Appendix A). 

The number of maxillary and mandibular aligners generated by each of the four software packages was also evaluated. The pairwise comparison showed significant differences between ClinCheck Pro and SureSmile (*p* ≤ 0.000), SureSmile and Ortho Insight 3D (*p* ≤ 0.014), SureSmile and Ortho Analyzer (*p* ≤ 0.009), and Ortho Insight 3D and Ortho Analyzer (*p* ≤ 0.000) for maxillary aligners (Table 4). The results varied slightly for mandibular aligners, with only ClinCheck Pro and Ortho Insight 3D (*p* ≤ 0.000), SureSmile and Ortho Insight 3D (*p* ≤ 0.000), and Ortho Insight 3D and Ortho Analyzer (*p* ≤ 0.000) exhibiting a significant difference (Table 5). Additionally, the number of attachments automatically generated by ClinCheck Pro and SureSmile was examined. The Kruskal–Wallis one-way analysis of variance demonstrated that ClinCheck Pro and SureSmile (*p* ≤ 0.000) differ significantly in the number of attachments produced (Table 6). Ortho Insight 3D and Ortho Analyzer were not included in this calculation since they are not capable of automatically generating attachments.

## 5. Discussion

Several software programs reported the ability to perform virtual tooth setups. Hou et al. studied the effect of digital diagnostic setups on orthodontic treatment. The records of six patients were obtained, and a total of twenty-two orthodontists and seven orthodontic residents were recruited for this study; moreover, a recommended treatment and another two alternative plans were obtained from each participant. A digital setup for each suggested treatment plan was made using the SureSmile software, and the confidence level in the success of their plans was recorded before and after viewing the digital setup. They found that viewing the digital setup resulted in changes to the treatment plans in about 24% of the cases, and using the digital setup was associated with higher levels of confidence in the selected plan [27].

In recent years, several companies have updated their software platforms to include virtual tooth setups. The aim of this study was to compare the differences in tooth movements when implementing the same virtual setup on four different software packages. To the best of our knowledge, this was the first study to do this type of comparison. The initial hypothesis was that, if we replicate virtual setups on four different software packages, the tooth movements found in each software will not significantly differ. If we used two different GPS applications to navigate from point one to point two, for instance, we would expect both applications to provide a similar route and estimated time of travel. This was not the case in our study, however. After implementing the same virtual setup in ClinCheck Pro, Ortho Analyzer, SureSmile, and Ortho Insight 3D, there were significant differences in the extrusion/intrusion and translation buccal and lingual movements. Due to these variations in tooth movements, the number of aligners and attachments automatically generated by each of the software packages also differed significantly.

One explanation for this could be that the different software packages use varying methods to segment the teeth and prepare the models. ClinCheck Pro and SureSmile use automated segmentation, which may explain why these two software packages do not significantly differ in any of the tooth movements. Ortho Analyzer and Ortho Insight 3D, in contrast, require manual segmentation in order to prepare the teeth for virtual tooth movement. Moreover, these manual segmentation procedures differ from one another. In Ortho Insight 3D, the steps include setting the facial axes, measuring the teeth, detecting landmarks, and aligning the roots with the crowns. Two studies tested the accuracy of predicting root inclinations and teeth long axes. Using the Ortho Insight 3D software, both studies concluded that root predictions cannot be considered accurate or reliable [28,29]. To overcome this problem, software programs now offer the ability to superimpose STL files on cone beam computed tomography (CBCT) images to increase the accuracy of roots’ positioning predictions using Ortho Analyzer; however, the steps include setting the mesial and distal points, defining cuts, and specifying the central axes of teeth.

A second explanation could be that the software packages use different centers of rotation when altering the inclination of teeth. ClinCheck Pro and SureSmile seem to rotate teeth around the approximated bone levels of healthy patients, around 1–2 mm apical to the cementoenamel junction (CEJ). This appears to resemble the motion of uncontrolled tipping. Ortho Insight 3D and Ortho Analyzer, on the other hand, seem to rotate teeth around the root apex. This more closely resembles the motion of controlled tipping. As an example, visualize proclining the mandibular incisors fifteen degrees via uncontrolled tipping. If you now visualize proclining the mandibular incisors fifteen degrees via controlled tipping, you will notice that a relative protrusive and intrusive effect is produced. In order to minimize these relative effects and mimic the uncontrolled tipping movement—as was the goal of this study—we must extrude and translate the teeth lingually. Extrusion/intrusion and translation buccal/lingual were the only movements found to be significantly different among the four software packages. The need for these compensatory movements may explain, at least in part, the differences that were observed in the tooth movement. 

## 6. Conclusions

There are statistically significant differences in extrusion/intrusion and translation buccal/lingual movements between software packages when progressing from the same initial malocclusion to the same final outcome. Similarly, the number of aligners and number of attachments differed significantly when implementing the same virtual setup on the software packages. In light of the growing popularity of in-house aligners and an upsurge in software packages that offer virtual setup options, clinicians should take these factors amongst others into consideration when choosing the appropriate software.

## Figures and Tables

**Figure 1 jcm-11-05351-f001:**
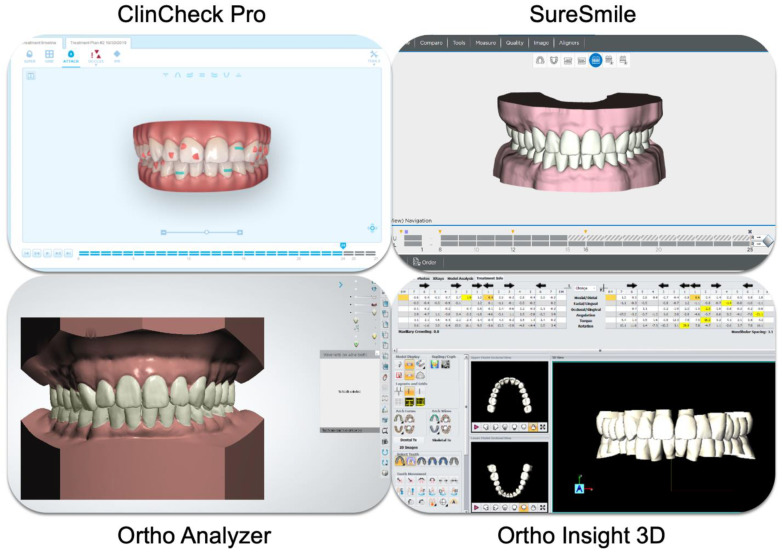
Replicated virtual setups from ClinCheck^®^ Pro, Ortho Analyzer^®^, SureSmile^®^, and Ortho Insight 3D^®^.

**Figure 2 jcm-11-05351-f002:**
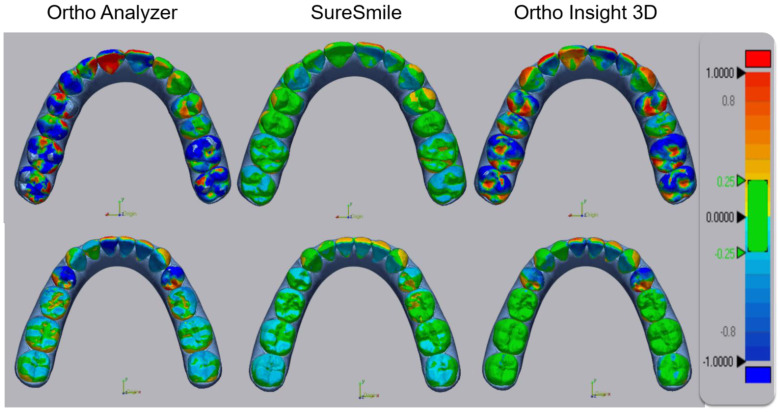
Color-coded maps outside the tolerance. Reference STL (ClinCheck^®^ Pro). Target STL (Ortho Analyzer^®^, SureSmile^®^, Ortho Insight 3D^®^).

**Figure 3 jcm-11-05351-f003:**
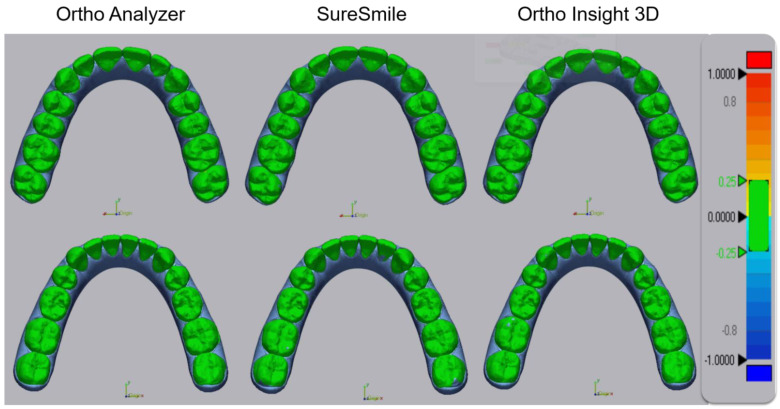
Color-coded maps inside the tolerance. Reference STL (ClinCheck^®^ Pro). Target STL (Ortho Analyzer^®^, SureSmile^®^, Ortho Insight 3D^®^).

**Figure 4 jcm-11-05351-f004:**
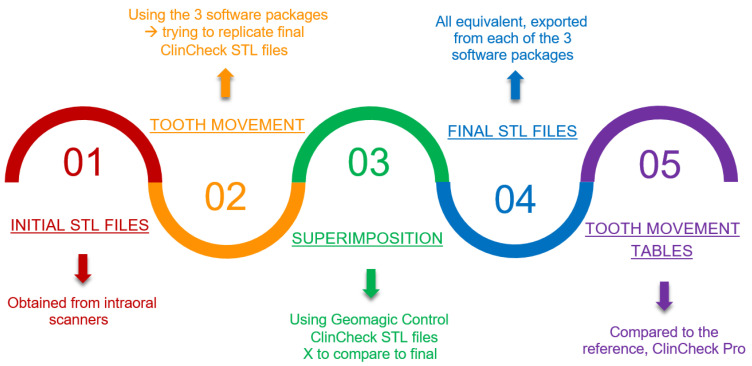
The workflow.

**Table 1 jcm-11-05351-t001:** Kruskal–Wallis one-way analysis of variance for all tooth movements.

Tooth Movement Variables	ClinCheck^®^ Pro (*n* = 682) Mean ± SD	Ortho Analyzer^®^ (*n* = 682) Mean ± SD	SureSmile^®^ (*n* = 682) Mean ± SD	Ortho Insight 3D^®^ (*n* = 682) Mean ± SD	*p*-Value
Extrusion/Intrusion (mm)	0.063 ± 0.025	0.329 ± 0.025	0.092 ± 0.025	0.136 ± 0.025	0.0001 *
Translation Buccal/Lingual (mm)	0.248 ± 0.036	0.077 ± 0.036	0.234 ± 0.036	0.133 ± 0.036	0.0004 *
Translation Mesial/Distal (mm)	0.016 ± 0.026	−0.027 ± 0.026	0.023 ± 0.026	0.023 ± 0.026	0.4630
Rotation Mesial/Distal (°)	−0.788 ± 0.350	−0.741 ± 0.350	−0.836 ± 0.350	−0.865 ± 0.350	0.9945
Angulation Mesial/Distal (°)	−0.149 ± 0.156	0.054 ± 0.156	−0.208 ± 0.156	−0.090 ± 0.156	0.6871
Inclination Buccal/Lingual (°)	0.740 ± 0.202	0.831 ± 0.202	0.686 ± 0.202	0.715 ± 0.202	0.9580

* indicates clinical significance based on *p* values of ≤0.05.

**Table 2 jcm-11-05351-t002:** Pairwise comparison for extrusion/intrusion.

Sample Comparison	Mean (mm)	Standard Deviation (mm)	*p*-Value
ClinCheck^®^ Pro–SureSmile^®^	0.063–0.092	±0.621–±0.652	0.226
ClinCheck^®^ Pro–Ortho Insight 3D^®^	0.063–0.136	±0.621–±0.637	0.026 *
ClinCheck^®^ Pro–Ortho Analyzer^®^	0.063–0.330	±0.621–±0.656	0.000 *
SureSmile^®^–Ortho Insight 3D^®^	0.092–0.136	±0.652–±0.637	0.311
SureSmile^®^–Ortho Analyzer^®^	0.092–0.330	±0.652–±0.656	0.000 *
Ortho Insight 3D^®^–Ortho Analyzer^®^	0.136–0.330	±0.637–±0.656	0.000 *

* indicates clinical significance based on *p* values of ≤0.05.

**Table 3 jcm-11-05351-t003:** Pairwise comparison for translation buccal/lingual.

Sample Comparison	Mean (mm)	Standard Deviation (mm)	*p*-Value
ClinCheck^®^ Pro–SureSmile^®^	0.248–0.234	±0.864–±0.832	0.814
ClinCheck^®^ Pro–Ortho Insight 3D^®^	0.248–0.133	±0.864–±0.915	0.004 *
ClinCheck^®^ Pro–Ortho Analyzer^®^	0.248–0.077	±0.864–±1.095	0.001 *
SureSmile^®^–Ortho Insight 3D^®^	0.234–0.133	±0.832–±0.915	0.009 *
SureSmile^®^–Ortho Analyzer^®^	0.234–0.077	±0.832–±1.095	0.002 *
Ortho Insight 3D^®^–Ortho Analyzer^®^	0.133–0.077	±0.915–±1.095	0.601

* indicates clinical significance based on *p* values of ≤0.05.

**Table 4 jcm-11-05351-t004:** Pairwise comparison for the number of maxillary aligners.

Sample Comparison	Mean (*n*)	Standard Deviation (*n*)	*p*-Value
ClinCheck^®^ Pro–SureSmile^®^	20.92–12.96	±6.64–±4.71	0.000 *
ClinCheck^®^ Pro–Ortho Insight 3D^®^	20.92–17.36	±6.64–±6.05	0.085
ClinCheck^®^ Pro–Ortho Analyzer^®^	20.92–25.76	±6.64–±12.01	0.379
SureSmile^®^–Ortho Insight 3D^®^	12.96–17.36	±4.71–±6.05	0.014 *
SureSmile^®^–Ortho Analyzer^®^	12.96–25.76	±4.71–±12.01	0.009 *
Ortho Insight 3D^®^–Ortho Analyzer^®^	17.36–25.76	±6.05–±12.01	0.000 *

* indicates clinical significance based on *p* values of ≤0.05.

**Table 5 jcm-11-05351-t005:** Pairwise comparison for number of mandibular aligners.

Sample Comparison	Mean (*n*)	Standard Deviation (*n*)	*p*-Value
ClinCheck^®^ Pro–SureSmile^®^	21.17–21.20	±5.20–±5.30	0.970
ClinCheck^®^ Pro–Ortho Insight 3D^®^	21.17–12.96	±5.20–±4.71	0.000 *
ClinCheck^®^ Pro–Ortho Analyzer^®^	21.17–25.76	±5.20–±12.01	0.461
SureSmile^®^–Ortho Insight 3D^®^	21.20–12.96	±5.30–±4.71	0.000 *
SureSmile^®^–Ortho Analyzer^®^	21.20–25.76	±5.30–±12.01	0.708
Ortho Insight 3D^®^–Ortho Analyzer^®^	12.96–25.76	±4.71–±12.01	0.000 *

* indicates clinical significance based on *p* values of ≤0.05.

**Table 6 jcm-11-05351-t006:** Kruskal–Wallis one-way analysis of variance for the number of attachments.

Sample Comparison	Mean (*n*)	Standard Deviation (*n*)	*p*-Value
ClinCheck^®^ Pro–SureSmile^®^	13.38–7.76	±3.09–±3.11	0.000 *

* indicates clinical significance based on *p* values of ≤0.05.

## Data Availability

The data underlying this article will be shared on reasonable request to the corresponding author.

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
