# Peer review of "Comparing 3D Tooth Movement When Implementing the Same Virtual Setup on Different Software Packages"

_jcm, 2022, doi:10.3390/jcm11185351_

Round 1

Reviewer 1 Report

Comments 0813

It is an interesting paper; however, major revision is suggested.

1.     Main text: please to change citation according to rule of “journal of Clinical Medicine”

2.     Introduction:

…. by Dr. Edward Angle and ………. by Zia Chishti and 41 Kelsey Wirth in 1997

- please to cite reference

…for CAD/CAM to be introduced to the field of dentistry by means of intraoral scanners (IOS). IOS were initially used in restorative dentistry to capture direct optical impressions of limited areas - please to cite reference

Recently, several companies have implemented CAD/CAM …..additional services including the digital fabrication of appliances, indirect bonding (IDB) trays and in-house aligners. - please to cite reference

3.     Materials & Methods:

please to give IRB number

STL - please to provide full name

all software packages and Geomagic Control X - please to provide detailed information, such as company name, location and country

4.     Discussion:

please to cite references and put into this portion   

please to provide your explanation about significantly difference in number of maxillary aligners and slightly different for mandibular aligners                                            

5.     Conclusion: which one software is better than others according your study ?                                                                                                                  

Author Response

We would like to thank the reviewers for the time and work they have put into our project, and for their patience.  We believe that we have addressed all concerns and we feel that our manuscript is now more clear and stronger than before.

Below are our answers

  1.  Main text: please to change citation according to rule of “journal of Clinical Medicine”

Thank you for pointing this out. Changes have been made.

2    Introduction:

…. by Dr. Edward Angle and ………. by Zia Chishti and 41 Kelsey Wirth in 1997

- please to cite reference

Thank you. A reference was added.

 for CAD/CAM to be introduced to the field of dentistry by means of intraoral scanners (IOS). IOS were initially used in restorative dentistry to capture direct optical impressions of limited areas - please to cite reference.

Thank you. A reference was added.

Recently, several companies have implemented CAD/CAM …..additional services including the digital fabrication of appliances, indirect bonding (IDB) trays and in-house aligners. - please to cite reference.

Thank you. A reference was added.

  1. Materials & Methods:

please to give IRB number

IRB number was added.

STL - please to provide full name

Full name was added to text

all software packages and Geomagic Control X - please to provide detailed information, such as company name, location and country

All details were added

  1. Discussion:

please to cite references and put into this portion   

please to provide your explanation about significantly difference in number of maxillary aligners and slightly different for mandibular aligners   

   Thank you for your suggestion. Authors mentioned two possible explanations for the differences between the software programs in the discussion section.                                             

  1. Conclusion: which one software is better than others according your study ?          

Thank you for your question. The purpose of this study was to compare the differences in tooth movements when implementing the same virtual setup on four different software Programs. There are software programs better than others on teeth segmentation and staging process for aligners from only an experience point of view. However, authors cannot make a strong statement of which software is better.

Reviewer 2 Report

This is very interesting research, but few aspects need clarification and there are some corrections that should be introduced.

1.     References in the text should be put in square brackets and be placed before the dot. 

2.     Line 49: clinicians should be started with lower case letter

3.     Line 92 – at what Department was this study conducted?

4.     You have described the exclusion criteria. Did you examined patients included in the study or only rely on their retrospective documentation? 

5.     How many researchers did the virtual setup? Did you check the repeatability?

6.     There is information in the text that the Figure 2 shows comparable results, within the tolerance but on the Figure 2 there is subtitle that those color-coded maps are outside the tolerance. Figure 1 do not show superimposition. The study design outline is on Figure 4. Please correct the list of Figures and references in the text.

7.     How big change of tooth position was planned in each software for a one splint?

8.     The discussion is very short. I understand that this is the first study comparing the setups and number of splints, but there were many studies before on aligners. Please discuss achieved results and compare them with current knowledge. I also feel that the introduction should also be improved. The short list of innovations at the beginning seems to be a bit out of place. Please focus more on technology than companies that are selling specific types of splints. 

9.     All products should include the information about the country and company in the brackets.

Author Response

We would like to thank the reviewers for the time and work they have put into our project, and for their patience.  We believe that we have addressed all concerns and we feel that our manuscript is now more clear and stronger than before.

Below are our answers in red:

  1. References in the text should be put in square brackets and be placed before the dot. 

Thank you. Changes have been made.

  1. Line 49: clinicians should be started with lower case letter

Thank you. Changes have been made

  1. Line 92 – at what Department was this study conducted?

Case Western Reserve University Department of Orthodontics.

  1. You have described the exclusion criteria. Did you examined patients included in the study or only rely on their retrospective documentation? 

This is a retrospective study based on existing data from the orthodontic department.

  1. How many researchers did the virtual setup? Did you check the repeatability?

All setups were done twice with the same researcher and another researcher redid 10 of the sample of interrater reliability.

  1. There is information in the text that the Figure 2 shows comparable results, within the tolerance but on the Figure 2 there is subtitle that those color-coded maps are outside the tolerance. Figure 1 do not show superimposition. The study design outline is on Figure 4. Please correct the list of Figures and references in the text.

Thank you, changes have been made.

  1. How big change of tooth position was planned in each software for a one splint?

Thank you for your question. Software program varies on staging per aligner. Some software programs use 0.25 per rotation or linear movement some other use o.5 for linear and 1.25 for rotation and therefore number of stages vary.

  1. The discussion is very short. I understand that this is the first study comparing the setups and number of splints, but there were many studies before on aligners. Please discuss achieved results and compare them with current knowledge. I also feel that the introduction should also be improved. The short list of innovations at the beginning seems to be a bit out of place. Please focus more on technology than companies that are selling specific types of splints. 

Changes have been made.

  1. All products should include the information about the country and company in the brackets

Thank you. Changes have been made.

Reviewer 3 Report

Dear Corresponding Authors,

Your manuscript is really interesting and well conducted.

Unfortunately it cannot be published in present form and it needs to be revised.

1)Please be sure to use only keywords accordingly to medical subject headings (Mesh word) for a better indexing.

2) First of all please add more background data in introduction section, furthermore, at the end of this section you should better state the main aim of the study. Maybe You could add a subparagraph called "Aim" or "objectives".

3) First, i ask you to check the plagiarism of your article using specific sites to get a similitary report (I recommend you grammarly pro: https://www.grammarly.com).

4)You need to review the grammar and English of your article, with the help of a native English speaker (you can specify who helped you in reviewing English in the acknowledgements) or using a site that can support you in English and grammar

5)About the Title of the article, I suggest you to modify it and add the type of article

6) Also I suggest you add a table with the list of abbreviations used in the text.

7) Add more recent references about the topic of the article (the most recent is dated 2019), dwelling in the introduction on articles published in 2022 and describing what your article will add compared to the last articles published; Preferably a published articles should be with 90 or more references, you don’t be afraid to add too many references. I suggest you some articles that will help you improve your article about the orthodontics movements and about the teledentistry and digital workflow (line 70)
Dento-Skeletal Class III Treatment with Mixed Anchored Palatal Expander: A Systematic Review DOI: 10.3390/app12094646
Application of vibrational spectroscopies in the qualitative analysis of gingival crevicular fluid and periodontal ligament during orthodontic tooth movement DOI 10.3390/jcm10071405 
Teledentistry in the Management of Patients with Dental and Temporomandibular Disorders Doi: https://doi.org/10.1155/2022/7091153 

8)Please expand conclusion section with main results and future perspectives of this study

Thank You,

Kind Regards

Author Response

We would like to thank the reviewers for the time and work they have put into our project, and for their patience.  We believe that we have addressed all concerns and we feel that our manuscript is now more clear and stronger than before.

Below are our answers in red:

1)Please be sure to use only keywords accordingly to medical subject headings (Mesh word) for a better indexing.

Thank you for your suggestion

2) First of all please add more background data in introduction section, furthermore, at the end of this section you should better state the main aim of the study. Maybe You could add a subparagraph called "Aim" or "objectives".

We have added a subparagraph where were explained the aim/objectives.

3) First, i ask you to check the plagiarism of your article using specific sites to get a similitary report (I recommend you grammarly pro: https://www.grammarly.com).

Thank you for your suggestion. Our search showed no similarity.

4)You need to review the grammar and English of your article, with the help of a native English speaker (you can specify who helped you in reviewing English in the acknowledgements) or using a site that can support you in English and grammar

A native English speaker from the authors reviewed the grammar and English.

5)About the Title of the article, I suggest you modify it and add the type of article

A change has been made.

6) Also I suggest you add a table with the list of abbreviations used in the text.

We have used 3 abbreviations. The table could look something like this. We are not sure where in the text it would be the most convenient to insert it.

CAD/CAM

Computer-aided design and manufacturing

IOS

Intraoral scanners

STL

Standard tessellation language

7) Add more recent references about the topic of the article (the most recent is dated 2019), dwelling in the introduction on articles published in 2022 and describing what your article will add compared to the last articles published; Preferably a published articles should be with 90 or more references, you don’t be afraid to add too many references. I suggest you some articles that will help you improve your article about the orthodontics movements and about the teledentistry and digital workflow (line 70)
Dento-Skeletal Class III Treatment with Mixed Anchored Palatal Expander: A Systematic Review DOI: 10.3390/app12094646
Application of vibrational spectroscopies in the qualitative analysis of gingival crevicular fluid and periodontal ligament during orthodontic tooth movement DOI 10.3390/jcm10071405 
Teledentistry in the Management of Patients with Dental and Temporomandibular Disorders Doi: https://doi.org/10.1155/2022/7091153 

We have updated the references

8)Please expand conclusion section with main results and future perspectives of this study

We have updated the conclusions

Round 2

Reviewer 1 Report

none